# Effect of Iron Depletion by Bloodletting vs. Observation on Oxidative Stress Biomarkers of Women with Functional Hyperandrogenism Taking a Combined Oral Contraceptive: A Randomized Clinical Trial

**DOI:** 10.3390/jcm11133864

**Published:** 2022-07-03

**Authors:** Manuel Luque-Ramírez, Andrés E. Ortiz-Flores, María Ángeles Martínez-García, María Insenser, Alejandra Quintero-Tobar, Sara De Lope Quiñones, Elena Fernández-Durán, María Lía Nattero-Chávez, Francisco Álvarez-Blasco, Héctor Francisco Escobar-Morreale

**Affiliations:** 1Diabetes, Obesity and Human Reproduction Research Group, Instituto Ramón y Cajal de Investigación Sanitaria (IRYCIS), 28034 Madrid, Spain; andreseduardo.ortiz@salud.madrid.org (A.E.O.-F.); manitamg@yahoo.es (M.Á.M.-G.); mariarosa.insenser@salud.madrid.org (M.I.); mariaalejandra.quintero@salud.madrid.org (A.Q.-T.); sara.lope@salud.madrid.org (S.D.L.Q.); elena.fernandezd@salud.madrid.org (E.F.-D.); marialia.nattero@salud.madrid.org (M.L.N.-C.); francisco.alvarez@salud.madrid.org (F.Á.-B.); hectorfrancisco.escobar@salud.madrid.org (H.F.E.-M.); 2Centro de Investigación Biomédica en Red Diabetes y Enfermedades Metabólicas Asociadas (CIBERDEM), Instituto de Salud Carlos III, 28034 Madrid, Spain; 3Department of Medicine and Medical Specialties, University of Alcalá, 28801 Alcalá de Henares, Spain; 4Department of Endocrinology and Nutrition, Hospital Universitario Ramón y Cajal, 28034 Madrid, Spain

**Keywords:** hyperandrogenism, iron, oxidative stress

## Abstract

Women with functional hyperandrogenism show both increased markers of oxidative stress and a mild iron overload. Combined oral contraceptives (COC) may worsen redox status in the general population. Since iron depletion ameliorates oxidative stress in other iron overload states, we aimed to address the changes in the redox status of these women as a consequence of COC therapy and of bloodletting, conducting a randomized, controlled, parallel, open-label clinical trial in 33 adult women with polycystic ovary syndrome or idiopathic hyperandrogenism. After three months of treatment with a COC, participants were randomized (1:1) to three scheduled bloodlettings or observation for another nine months. After taking a COC, participants showed a mild decrease in their plasma electrochemical antioxidant capacity, considering fast-acting antioxidants [MD: −1.51 (−2.43 to −0.60) μC, *p* = 0.002], and slow-acting antioxidants [MD: −1.90 (−2.66 to −1.14) μC, *p* < 0.001]. Women submitted to bloodletting showed a decrease in their non-enzymatic antioxidant capacity levels (NEAC) throughout the trial, whereas those individuals in the control arm showed a mild increase in these levels at the end of the study (Wilks’ λ: 0.802, F: 3.572, *p* = 0.041). Decreasing ferritin and plasma hemoglobin during the trial were associated with worse NEAC levels. COC may impair redox status in women with functional hyperandrogenism. Decreasing iron stores by scheduled bloodletting does not override this impairment.

## 1. Introduction

Functional hyperandrogenism—including polycystic ovary syndrome (PCOS) and hyperandrogenic-related disorders—is a prevalent disorder in premenopausal women worldwide [1]. The development of oxidative stress derives from an unbalance between reactive oxygen (ROS) and nitrogen species generation, and antioxidant mechanisms [2]. ROS are produced in many cellular compartments as a part of normal cell activity [3]. Even though the leakage of electrons in the mitochondrial respiratory chain is the main source of ROS as by-products of nutrients intake, an excessive amount of ROS can arise from other processes such as excessive stimulation of NAD(P)H oxidases, enzymatic activation of cytochrome p450 at microsome and mitochondria, transition metal ions such as iron, peroxisome activity, alterations in the endoplasmic reticulum folding pathway, or thymidine and polyamine catabolism [3,4,5]. Oxidative stress induced by mitochondrial dysfunction participates in the pathogenesis of PCOS and its related cardio-metabolic complications [2,6]. Among the facilitators of oxidative stress in these women, hyperandrogenism, insulin resistance (IR), adiposity, and chronic inflammation promote mitochondria-mediated damage and reactive species production [7,8,9,10]. In conceptual agreement, macronutrient-induced oxidative stress modulated by those factors has been reported in women with PCOS [11,12,13]. Closing a vicious circle, oxidative stress reciprocally promotes androgen excess by enhancing 17,20-lyase activity through P450c17 phosphorylation [14], and increases the bioavailability of circulating testosterone by reducing sex hormone-binding globulin expression and secretion [15].

An increase in iron tissue depots might also contribute to oxidative stress in hyperandrogenic women. Patients with functional hyperandrogenism show a mild iron overload—as defined by increased circulating ferritin levels—compared to women without androgen excess of similar age and body mass index, even though their ferritin concentrations may remain within the normal range [16,17,18]. Iron is a strong pro-oxidant molecule and is implicated in the pathogenesis of chronic metabolic diseases [19]. In conceptual agreement, iron depletion by phlebotomy or nutritional counseling improves several metabolic endpoints not only in acquired iron overload states but also in persons with normal iron stores [20]. On the other hand, reduced iron stores, such as those of frequent blood donors, link to incident/prevalent cardiovascular disease in population studies [21]. Underlying this finding may be that ferritin acts as a major antioxidant molecule, protecting the endothelium from oxidative damage [22]. Furthermore, iron deficiency also impairs mitochondrial activity [23].

In women with functional hyperandrogenism not seeking pregnancy, combined oral contraceptives (COC) are the mainstay of treatment for hyperandrogenic dermo-cosmetic complaints while providing endometrial protection. After extensive first-pass intestinal and hepatic metabolism [24], the estrogenic —most commonly, 17α-ethinylestradiol (EE)—and progestogenic compounds of oral contraceptives undergo phase 1 and 2 metabolic reactions to be mainly excreted in feces as glucuronide conjugates [25]. The use of COC may affect the balance between phase 1 and 2 biotransformations facilitating oxidative stress [26,27]. Furthermore, the ferric reducing ability of plasma—a measure of antioxidant capacity—appears to be reduced in COC users [28]. Hence, women with functional hyperandrogenism taking COC become an excellent human model to test the possible effects of iron depletion on redox status.

We hypothesized that: (i) COC use may lead to increase oxidative stress biomarkers in women with functional hyperandrogenism, and (ii) iron depletion by bloodletting could counteract this unwanted effect of COC. Thus, this study aimed to describe the changes in the electrochemical antioxidant capacity, lipid peroxidation, reactive nitrogen species-induced nitrative stress, and plasma non-enzymatic antioxidant capacity (NEAC) in women with functional hyperandrogenism after starting standard treatment with COC, and the effects on redox status of iron depletion resulting from repeated phlebotomies.

## 2. Materials and Methods

### 2.1. Study Design

This preplanned study derived from a parallel and controlled non-commercial randomized clinical trial (RCT) previously reported [29]. We registered the study protocol at ClinicalTrials.gov (Identifier: NCT02460445. Date of first registration: 2 June 2015).

### 2.2. Patients

The study’s participants were premenopausal women with functional hyperandrogenism—including hyperandrogenic PCOS and idiopathic hyperandrogenism—consecutively recruited at the Reproductive Endocrinology clinic from an Academic Hospital in Madrid, Spain. We based the diagnosis of hyperandrogenic PCOS on the presence of clinical (i.e., hirsutism) and/or biochemical hyperandrogenism plus ovulatory dysfunction and/or polycystic ovarian morphology by sonographic assessment [30]. Idiopathic hyperandrogenism was defined by the presence of clinical and biochemical hyperandrogenism, in the absence of ovulatory dysfunction and polycystic ovarian morphology [31].

We described in detail other inclusion and exclusion criteria elsewhere [29]. To be included in the study, none of the women had a prior history of dyslipidemia, hypertension, prediabetes, diabetes mellitus, gestational diabetes, or cardiovascular events, nor had been treated with COC, antiandrogens, insulin sensitizers, or any drug that might interfere with blood pressure regulation, lipid profile, or carbohydrate metabolism, or oral/parenteral iron therapy, for the previous 3 months. From June 2015 to June 2019, we screened 63 consecutive women, although 26 of them did not participate in the study because of different reasons. We randomized the remaining 37 women, but four of them did not complete the run-in period, being excluded from later analyses. Finally, 33 women entered the trial, with 26 of them completing the study (Figure 1).

### 2.3. Randomization

We used stratified block randomization to allocate the patients (1:1) to scheduled bloodletting or to observation. Blocks of 10 sealed opaque envelopes (5 per arm) served for treatment assignment. One investigator (M.L.-R.) generated the randomization envelopes. Then, another researcher (A.E.O.-F.) enrolled and assigned the participants to their arm of treatment. The laboratory personnel in charge of biochemical assays were blinded to both women’s features and treatment allocation. We did not use other masking methods after randomization.

### 2.4. Intervention

After randomization (month −3 visit), participants started a run-in period of treatment with a COC [21 pills containing 35 μg of EE plus 2 mg of cyproterone acetate (CPA) followed by 7 placebo pills, Diane^35^ Diario; Schering España S.A., Madrid, Spain]. After 3 cycles of treatment (month 0 visit), the bloodletting intervention began in the experimental arm, consisting of 3 scheduled phlebotomies during a 9-month period, with study visits at months 6 and 9 (end of the study) (Figure 1). Participants in both arms of the RCT maintained their treatment with COC throughout all study visits.

### 2.5. Outcomes Assessment

We described the RCT procedures elsewhere [29]. Clinical, anthropometric, and biochemical evaluations have been previously reported [29]. Study subjects completed a baseline assessment (month −3 visit) including an assessment of oxidative stress biomarkers. We re-evaluated participants’ redox status at months 0, 6, and 9. At the month −3 visit, we performed these evaluations between days 3 and 9 of spontaneous menstrual bleeding or after excluding pregnancy in amenorrheic patients. We performed the same protocol at months 0, 6, and 9 visits regardless of the day of the menstrual cycle because all women were on the same COC.

### 2.6. Biochemical and Hormonal Assays

The assays used for biochemical and hormonal phenotyping have been already reported [29]. For oxidative stress analyses, we obtained plasma samples after 12-h overnight fasting and immediately frozen aliquots at −80 °C until thawed and assayed. In the present study, samples previously stored for electrochemical antioxidant capacity and plasma NEAC measurements were missing in one of the patients allocated to the control group. We assayed all plasma samples by duplicate, using, for each assay, kits from the same production lot.

We directly measured electrochemical antioxidant capacity in 60 mL of plasma at room temperature. This measurement was based on the redox potential readings obtained from a potentiostat (e-BQC lab, BIOQUOCHEM S.L., Llanera-Asturias, Spain), and expressed in charge units [micro-Coulombs (μC)]. Electrochemical antioxidant capacity represents the charge of the electrons released by the antioxidants present in plasma to neutralize free radicals. This device distinguishes between compounds of high (Q1) or low rate (Q2) of free radical scavenging and also reports the sum of both (QT), with an inter-assay coefficient of variation (CV) of 8%.

We assayed plasma thiobarbituric acid reactive substances (TBARS) concentrations by a commercial colorimetric method (TBARS Assay Kit, Cayman Chemical, Ann Arbor, MI) with intra- and inter-assay CV of 5.5 and 5.9%, respectively. The TBARS assay expresses the results in terms of μM of malondialdehyde (MDA) equivalents per liter since water solutions of MDA at different concentrations are used to prepare the standard curve. Plasma 2-nytrotirosine concentrations were determined by a competitive ELISA kit (OxiSelect™ Nitrotyrosine ELISA Kit, Cell Biolabs, Inc., San Diego, CA, USA) by comparing its absorbance with that of a known bovine serum albumin nitrated curve, with a detection sensitivity range of 20 nM to 8.0 μM, and intra- and inter-assay CV of 6 and 10%, respectively. Lastly, we used a commercial colorimetric method (Antioxidant Assay Kit, Cayman Chemical, Ann Arbor, MI, USA) to measure NEAC. The assay relies on the ability of antioxidants in the plasma to inhibit the oxidation of 2,2′-Azino-di-(3-ethylbenzthiazoline sulphonate) by metmyoglobin. NEAC of the sample is compared with that of Trolox, a water-soluble tocopherol analog, and expressed as mM Trolox equivalents. This assay has a detection sensitivity range between 0.044 and 0.330 mM, and intra- and inter-assay CV of 3 and 3.4%, respectively.

### 2.7. Power Calculation

We previously calculated sample size calculations for the primary outcomes of the RCT [29]. In the present study, we used the free online application provided by the Program of Research in Inflammatory and Cardiovascular Disorders from the Institut Municipal d’Investigació Mèdica, Barcelona, Spain [www.imim.cat/ofertadeserveis/software-public/granmo/ (accesed on 22 July 2021)] for power calculations. During the experimental phase of the trial, the inclusion of 16 (or 15 in the case of electrochemical antioxidant capacity and NEAC) and 17 women by intention-to-treat (ITT) analysis in each arm of treatment provided a power (1-β) of 80% —at a two-side 0.05 significance level— to detect a minimum mean of the differences (MD) between arms of treatment of 0.8 µC, 0.7 μC, 1.5 μC, 0.14 μM, 16 mM, and 0.2 μM in the change of Q1, Q2, QT, TBARS, 2-nytrotirosine, and NEAC values, respectively. These calculations use a common standard deviation (SD) of 0.8 μC, 0.9 μC, 1.6 μC, 0.16 μM, 28 mM, and 0.2 μM, respectively, for these measures, and correlation coefficients between the visit months 0 and 9 (end of the study) for these measurements of 0.457, 0.658, 0.546, 0.597, 0.822, and 0.377, respectively, as those observed in the RCT.

### 2.8. Statistical Analysis

Data are shown as means ± SD, MD, or counts (percentage) unless otherwise stated, with their respective 95% confidence intervals (CI) (lower limit to upper limit). For continuous variables, we assessed normality using the Kolmogorov–Smirnov test. We applied logarithmic or two-step transformations [32] to ensure normality as needed. To assess the effect of COC administration during the run-in period, we applied paired *t*-tests.

To address the effect of bloodletting on oxidative stress markers at each visit, we implemented a repeated-measures general linear model (GLM) including the arm of treatment as the between-subjects effect and the visit (months 0, 6, and 9) as the within-subjects effect. A statistically significant interaction of the between- and within-effects would indicate that the changes were different depending on the arm of treatment. We used Mauchly’s test to measure sphericity, and then applied Greenhouse–Geisser epsilon adjustment as needed. In pairwise comparisons, we dealt with multiplicity by adjusting CI using Bonferroni’s method. We also explored the changes in oxidative stress biomarkers throughout the RCT as areas under the curve (AUC) were determined using the trapezoidal rule. We compared the resulting AUC in each arm of treatment by unpaired *t*-tests.

We conducted all inferential statistics by ITT. Participants randomized and completed the run-in period (from month −3 to 0), and comprised the ITT group, regardless of whether or not they completed the RCT. ITT analyses assumed that the dependent variables had not changed at the missing visits with respect to the values observed in the previous visit.

A nominal two-sided α level was set at 0.05. We performed statistical analyses using PASW Statistics 18 (IBM España S.A., Madrid, Spain).

## 3. Results

### 3.1. Short-Term Effects of Combined Oral Contraceptives on Oxidative Stress Biomarkers

The baseline characteristics of the study participants are shown in Table 1. After 3 months on COC (run-in period), participants considered as a whole had a significant decrease in their plasma electrochemical antioxidant capacity, both in Q1 values [MD: −1.51 (−2.43 to −0.60) μC, *p* = 0.002], Q2 values [MD: −1.90 (−2.66 to −1.14) μC, *p* < 0.001], as in QT values [MD: −3.40 (−5.02 to −1.78) μC, *p* < 0.001] (Figure 2). Furthermore, 24 (77.4%) and 26 (83.9%) women decreased Q1 and Q2 values, respectively, during this period of the trial. These findings were virtually identical in those women with (42%) or without (58%) obesity, defined by a body mass index (BMI) ≥ 30 kg/m^2^. We did not observe any significant change in TBARS, 2-nitrotyrosine, or NEAC (Figure 2).

### 3.2. Effect of Iron Depletion by Bloodletting on Oxidative Stress Biomarkers

As expected, scheduled phlebotomies were effective to decrease iron depots in those patients allocated to the intervention arm of the study (Figure 3) [29]. We did not observe any significant change in oxidative stress biomarkers during the experimental phase of the study, considering all participants as a whole (Figure 4). There was a significant interaction between the visit and arm of follow-up on NEAC values (Wilks’ λ: 0.802, F: 3.572, *p* = 0.041, η^2^_p_: 0.198, Figure 4). That interaction consisted of women submitted to bloodletting showing a mild decrease in their NEAC over the trial, whereas those women allocated to the control arm had a small increase in their concentrations from visit month 6 to the end of the study (Figure 4). There were no statistically significant differences between the arms of the study on that measurement in the adjusted pairwise comparisons derived from repeated-measures GLM [MD: −0.13 (−0.55 to 0.30), *p* = 0.478] or in the AUC of NEAC [difference of the means: 0.99 (−1.84 to 3.81), *p* = 0.519]. The introduction of obesity or BMI as a between-subjects factor or covariate in the repeated-measures GLM models did not change any abovementioned finding.

### 3.3. Determinants of Changes in Oxidative Stress Biomarkers

The changes in anthropometric, hormonal, metabolic, ferrikinetic, and hematimetric variables throughout both the run-in and experimental phases of the trial have been reported already [29]. In Table 2, we show correlation analyses between the changes in markers of electrochemical antioxidant capacity and those observed in anthropometric, metabolic, hormonal, and ferrokinetics variables. During the run-in period, the individual changes in Q1 and Q2 values inversely correlated with the changes in plasma hemoglobin values and transferrin saturation, respectively, even though only mean plasma hemoglobin showed a significant decrease during this period of the trial [29]. In the same line, the individual changes in QT values after 3 months of treatment with COC negatively correlated with the changes in the homeostasis model assessment of IR (HOMA-IR) (Table 2), despite mean HOMA-IR values did not show any statistically significant change when all participants were considered as a whole [MD: −0.15 (−0.99 to 0.69)].

We did not observe any significant correlation between the changes in these variables and oxidative stress biomarkers during the experimental phase of the trial in any arm of the trial (data not shown). However, a repeated-measures GLM model introducing the study visit as a within-subjects factor, and ferrokinetics and hematimetric variables grouped by tertiles showed that the AUCs of ferritin and plasma hemoglobin interacted with the study visit on NEAC values (Wilks’ λ: 0.779, F: 4.112, *p* = 0.027, η^2^_p_: 0.221; and F: 4.322, *p* = 0.022, η^2^_p_: 0.126, respectively, Figure 5). We did not find other statistically significant associations.

## 4. Discussion

We hereby show that a COC comprised of EE and CPA worsens the antioxidant capacity of women with functional hyperandrogenism soon after beginning treatment. As stated, this deleterious effect is directly associated with changes in transferrin saturation, plasma hemoglobin, and IR. However, iron depletion by scheduled bloodletting did not improve the redox status of these women. Furthermore, our finding suggests that iron deficiency—as defined by low ferritin concentrations induced by repeated phlebotomies or by means of restoring regular menses after taking COC—may counteract any antioxidant compensatory response.

Women with functional hyperandrogenism have increased markers of oxidative stress at fasting [2,33], besides a pro-oxidant response to glucose [12,34] or saturated-fat ingestion [11]. Cellular ROS generation linked to a disrupted mitochondria DNA replication and oxidative phosphorylation efficiency, in addition to low levels of antioxidant enzymes, appear to be responsible for the redox imbalance in these patients [2,35]. Typical features of PCOS and its related hyperandrogenic disorders, such as obesity and abdominal adiposity [36], hyperandrogenemia [37], IR and compensatory hyperinsulinism [6], chronic subclinical inflammation [10], and/or inherited genetic variants [35], may impair mitochondrial oxidative metabolism and reduce antioxidant defenses.

In this scenario, the effect of COC—and their different compounds—on the redox status of hyperandrogenic women becomes an important topic. Although hormonal replacement therapy may prevent oxidative damage in postmenopausal women, COC increases oxidative status in young users from the general population [27,38]. Cytochrome P450 1A2 activity is inhibited by COC, whereas enzymes involved in phase 2 biotransformations are induced in COC users [28]. Therefore, hydroxyl radicals from phase 1 metabolic activity would not be a major contributor to oxidative stress in COC users. Interestingly, the estrogen-induced phase 2 enzyme glycine *N*-acyltransferase increases glycine consumption [39]. Glycine reduces superoxide anion radical release, protein carbonyl, and lipid peroxidation, and on the contrary, increases the biosynthesis of glutathione (GSH) [40]. This putative mechanism for COC-induced oxidative stress agrees with the changes observed in the metabolome of COC users [39].

Evidence about the effect of hormonal contraceptives on the redox status of hyperandrogenic women is lacking. We found a mild decrease in the electrochemical antioxidant capacity of hyperandrogenic women after beginning a COC comprised of EE and CPA, a deleterious effect maintained throughout the 12 months of the study. However, we did not identify any effect attributable to the COC pill on lipid peroxidation, reactive nitrogen species-induced nitrative stress, or NEAC measured by a classic colorimetric assay. Plasma antioxidant capacity derives from multiple processes and circulating metabolites. Thus, NEAC assays are sensitive to a particular combination of antioxidants but exclude others, and the correlation between different methods is poor [41].

Interestingly, electrochemical antioxidant capacity is able to measure thiol components, and therefore GSH, unlike the Trolox Equivalent Antioxidant Capacity used in the present study [42]. A decrease in GSH content in leukocytes of women with PCOS at fasting is linked to the development of IR and hyperandrogenism [8]. Decreased GSH concentrations are reported in young healthy women using COC [43]. In this vein, we previously communicated that a COC containing EE plus CPA increases circulating homocysteine concentrations in non-smoking women with PCOS [44]. The metalloproteins (with heme moiety)-induced auto-oxidation of homocysteine thiol groups forming disulfides or with free cysteine residues of proteins results in the generation of ROS, and proteins with impaired functions such as hemoglobin, that lose their reduced state [45]. This hypothesis, although merely speculative, would be in consonance with the observed changes induced by the COC in the electrochemical antioxidant capacity of our study participants, and the inverse correlation with the changes in the HOMA-IR index, plasma hemoglobin, and transferrin saturation index over the same period of the trial.

A mild iron excess state would be also able to contribute to oxidative stress damage in individuals with functional hyperandrogenism [18,46]. In pernicious crosstalk, oxidative stress promotes hyperandrogenism [14], reproductive dysfunction [47], or insulin resistance [34]. Particularly, iron-induced oxidative damage impairs insulin secretion, decrease insulin clearance, and muscle glucose uptake [19,48]. In keeping, a mild iron overload is especially evident in women with functional hyperandrogenism and abnormal glucose tolerance [17]. Unfortunately, iron depletion by repeated bloodletting was not accompanied by any improvement in the redox status of the normorferritinemic women with functional hyperandrogenism included in our RCT. Intensive iron depletion by phlebotomy does attenuate oxidative stress in subjects with hereditary hemochromatosis [49,50], although pro-oxidant biomarkers may remain elevated during the chronic maintenance phase of phlebotomy therapy [50]. In addition to restoring systemic iron homeostasis, phlebotomy upregulates intracellular manganese superoxide dismutase, and reduces nuclear DNA damage in peripheral blood mononuclear cells in these patients [51]. Iron reduction therapy by phlebotomy also reduces lipid peroxidation and oxidative stress in patients with chronic hepatitis C and increased liver iron depots [52]. However, a reduction of iron stores does not significantly improve biochemical markers of oxidative stress, such as cytochrome P450 2E1 activity or MDA levels, in the dysmetabolic iron overload syndrome [53].

In our study, participants had ferritin levels within a normal range and no evidence of liver iron overload; decreasing tissue iron depots by phlebotomy did not restore the electrochemical antioxidant capacity to the levels observed before beginning COC treatment, or induced any improvement in TBARS, 2-nitrotyrosine, or NEAC. Even more, we observed a statistically significant increase in NEAC at the end of the study in the women allocated to the control arm, and this change only occurred in those participants who did not reduce, or increase, their circulating ferritin concentrations over the RCT.

Excessive iron depletion may blunt this potential counter-regulatory NEAC response by several mechanisms. Being a serum indicator of tissue iron depots, ferritin also has antioxidant properties—attributable to either iron storage and/or the intrinsic ferroxidase of its heavy subunit—participating in the antioxidant against iron-induced oxidative damage [22]. Lower circulating ferritin levels may reduce the plasma’s ability to inhibit the oxidation of 2,2′-Azino-di-(3-ethylbenzthiazoline sulphonate) by metmyoglobin, which is the antioxidant capacity that the NEAC assay relies on. In addition, most women allocated in the intervention arm of the study presented with low ferritin concentrations and/or transferrin saturation percentage, and in some cases, low hemoglobin, or hematocrit values [29]. Iron deficiency anemia is associated with oxidative stress by increasing hemoglobin auto-oxidation and generation of ROS [54].

Obviously, our study has several limitations. In all antioxidant research, the main weakness derives from taking a picture of a dynamic process such as redox homeostasis in a sample removed from its biological context. Although we used a battery of assays to characterize the fasting redox status of our study participants, the radical scavenging observed in vitro, using an arbitrarily selected oxidant generator, does not necessarily resemble that occurred in vivo, which is characterized by the enzymatic maintenance of a steady state [55]. Even though we followed a strict protocol for storing samples, and all of them were assayed by duplicate using kits from the same lot provided by the manufacturer, we cannot discard the impact of storage conditions on oxidative stress biomarkers. In the same line, measurement of products of lipid peroxidation and reactive nitrogen species also has important hurdles [56,57], including a significant intra-individual variability [58]. Moreover, the TBARS assay used for lipid peroxidation may be relatively unspecific [59]. Another limitation inherent to our study design is a small sample size. Thus, we might have overlooked small outcome differences between subgroups. Finally, yet importantly, women with functional hyperandrogenism only show a modest increase in body iron stores. Evidence for the beneficial effect of bloodletting mostly comes from conditions with an important burden of tissue iron excess. In the absence of other iron excess disorders, ferritin levels are within normal ranges in most of these young hyperandrogenic women. There is no evidence for increased liver depots in these patients either. Furthermore, we excluded women with known metabolic comorbidities before their study inclusion. In short, a possibility exists that the inclusion of hyperandrogenic women with more severe metabolic phenotypes or presenting with more severe iron overload could have been more likely to obtain a benefit from iron depletion by scheduled phlebotomies.

## 5. Conclusions

Women with functional hyperandrogenism and circulating ferritin within the normal range suffer from a mild decrease in their electrochemical antioxidant activity after beginning conventional treatment with an antiandrogenic COC. This derangement is associated with modest changes in ferrokinetics and insulin sensitivity. Iron depletion by bloodletting does not have a significant impact on the products of lipid peroxidation, reactive nitrogen species, or electrochemical antioxidant capacity. On the contrary, decreasing ferritin levels might be associated with a poorer NEAC. Despite a mild iron overload playing a role in the pathogenesis of functional hyperandrogenism and its metabolic comorbidities, we cannot recommend iron depletion for most women with this condition in view of current evidence. The effect of COC treatment on the redox status of these women merits further consideration. Whether or not iron depletion by bloodletting may be beneficial for hyperandrogenic women with hyperferritinemia and other metabolic morbidities, taking or not COC, would need further research.

## Figures and Tables

**Figure 1 jcm-11-03864-f001:**
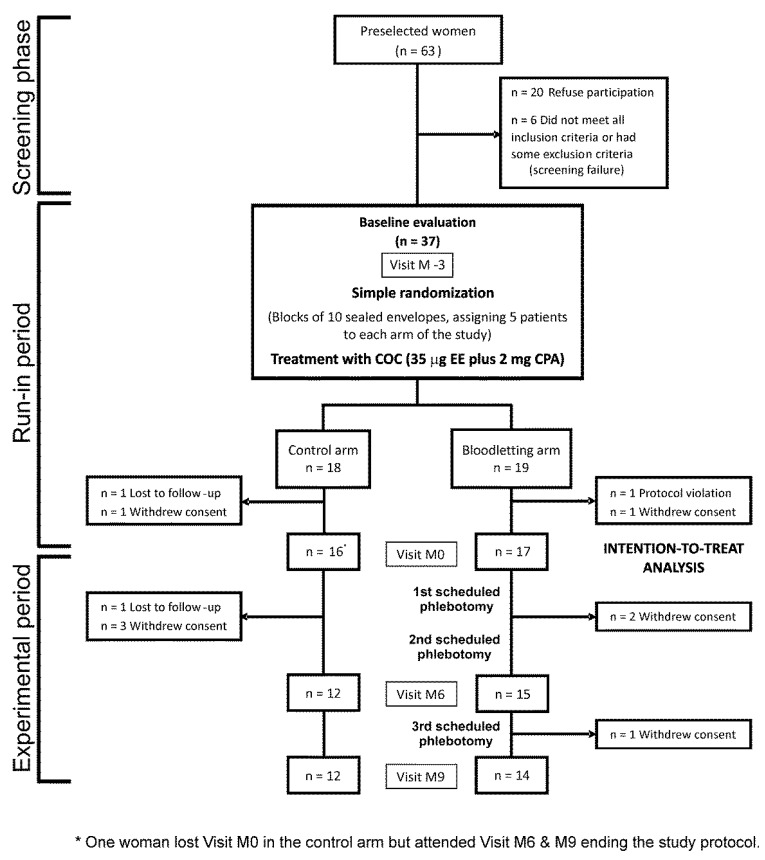
Flow chart of the study. The figure includes the numbers of participants randomly assigned to the arms of the study, those receiving the intended intervention, and the losses and exclusions, together with their reasons, occurring after randomization.

**Figure 2 jcm-11-03864-f002:**
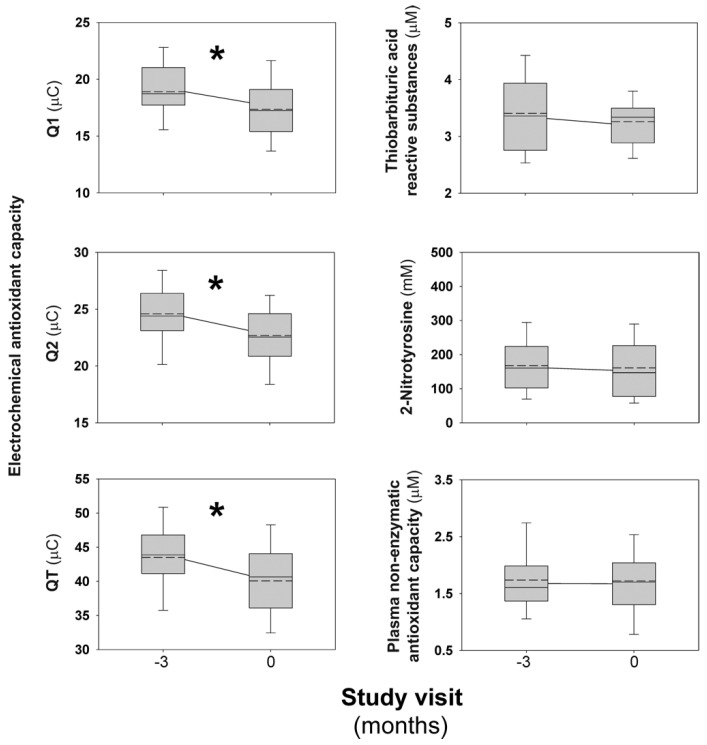
Short-term effects of combined oral contraceptive on oxidative stress biomarkers. The box indicates the 25th and 75th percentiles, the solid and short dashed lines within the box mark the median and mean, respectively. Whiskers below and above the box indicate the 10th and 90th percentiles. * Statistically significant change in mean values from month −3 to 0.

**Figure 3 jcm-11-03864-f003:**
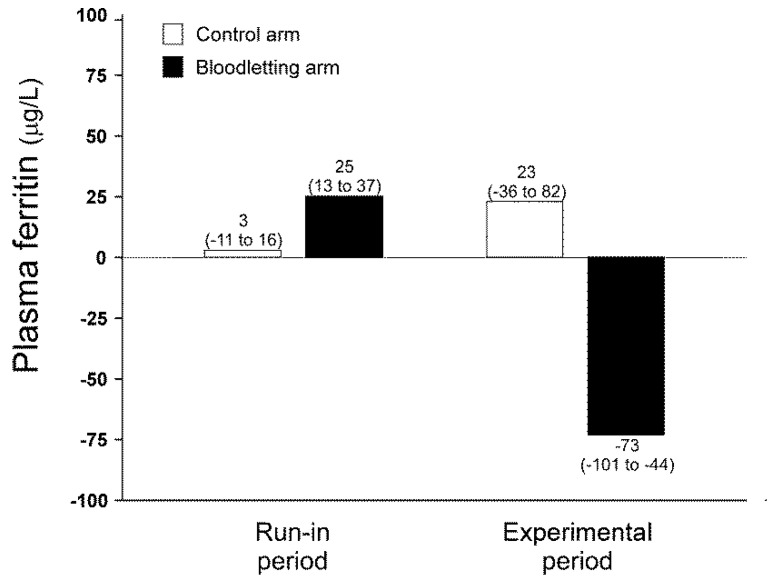
Changes in circulating ferritin in each arm of the trial throughout the study. Data are shown as mean of the differences and 95% CI.

**Figure 4 jcm-11-03864-f004:**
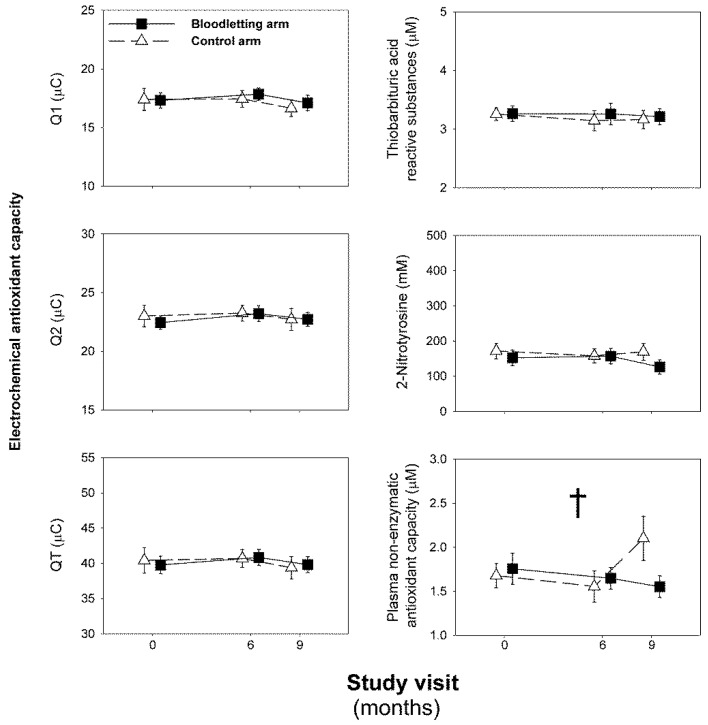
Changes in oxidative stress biomarkers throughout the experimental phase of the trial. We show data as means (SEM) of the patients remaining at each visit of the trial, even though we conducted intention-to-treat statistical analyses. Data were submitted to a repeated-measures general linear model. † Statistically significant interaction between the visits and the arm of the study.

**Figure 5 jcm-11-03864-f005:**
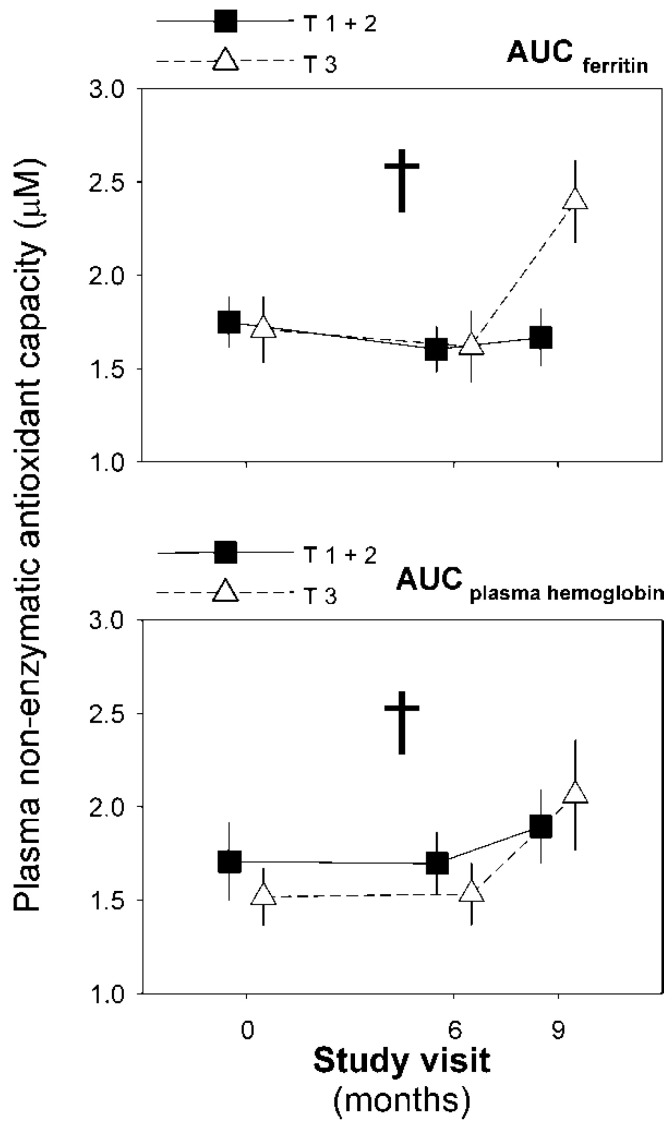
Changes in plasma non-enzymatic antioxidant activity throughout the experimental phase of the trial as a function of the changes in plasma ferritin and hemoglobin grouped by tertiles. We show data as means (SEM) of the patients remaining at each visit of the trial even though we conducted intention-to-treat statistical analyses. Data were submitted to a repeated-measures general linear model introducing the upper tertile (T3) vs. mid and lower tertiles (T1 + 2) as between-subjects factors. † Statistically significant interaction between the visits and tertile subgroups.

**Table 1 jcm-11-03864-t001:** Baseline characteristics and fasting plasma redox status of the patients with functional hyperandrogenism (PCOS or idiopathic hyperandrogenism) randomized to treatment with a combined oral contraceptive pill plus scheduled phlebotomies (experimental arm) or to COC plus observation (control arm).

	Intention-to-Treat Population		
	All Women	Bloodletting Arm	Control Arm
	(*n* = 33)	(*n* = 17)	(*n* = 16)
*Age (*years*)*	25 ± 6	25 ± 7	25 ± 6
	(23 to 27)	(21 to 29)	(22 to 28)
*Body mass index (*kg/m^2^*)*	28.9 ± 8.0	29.6 ± 8.1	28.3 ± 8.1
	(26.1 to 31.7)	(25.4 to 33.8)	(24.0 to 32.6)
*Waist circumference (*cm*)*	89 ± 18	90 ± 16	87 ± 20
	(83 to 95)	(82 to 98)	(76 to 98)
*Frequency of obesity, n (%)*	14 (42)	8 (47)	6 (38)
	(27 to 59)	(26 to 69)	(19 to 61)
*Total testosterone (*nmol/L*)*	2.75 ± 1.04	2.6 ± 1.1	2.9 ± 1.0
	(2.4 to 3.1)	(2.1 to 3.2)	(2.3 to 3.4)
*Calculated free testosterone (*pmol/L*)*	52 ± 22	52 ± 25	51 ± 20
	(44 to 60)	(39 to 65)	(40 to 62)
*Dehydroepiandrosterone-sulphate (*μmol/L*)*	7.56 ± 2.92	7.3 ± 2.2	7.8 ± 3.9
	(6.53 to 8.60)	(6.2 to 8.5)	(5.7 to 9.9)
*Homeostasis model assessment of insulin resistance*	2.8 ± 2.5 (1.9 to 3.6)	2.7 ± 2.5 (1.4 to 4.0)	2.8 ± 2.5 (1.5 to 4.1)
*Electrochemical antioxidant capacity (*μC*)*			
*Q1*	18.9 ± 2.6	19.3 ± 2.2	18.5 ± 3.0
	(18.0 to 19.8)	(18.1 to 20.4)	(16.8 to 20.2)
*Q2*	24.6 ± 2.7	24.9 ± 1.8	24.3 ± 3.4
	(23.6 to 25.6)	(24.0 to 25.8)	(22.4 to 26.2)
*QT*	43.5 ± 5.0	44.2 ± 3.4	42.7 ± 6.3
	(41.7 to 45.3)	(42.5 to 46.0)	(39.2 to 46.2)
*Thiobarbituric acid reactive substances (*μM*)*	3.40 ± 0.69	3.35 ± 0.68	3.47 ± 0.72
	(3.16 to 3.65)	(3.00 to 3.70)	(3.09 to 3.85)
*2-Nitrotyrosine (*mM*)*	168 ± 83	166 ± 86	169 ± 83
	(138 to 197)	(122 to 211)	(125 to 213)
*Non-enzymatic antioxidant capacity (*μM*)*	1.74 ± 0.57	1.71 ± 0.55	1.77 ± 0.60
	(1.54 to 1.94)	(1.43 to 1.99)	(1.44 to 2.10)

Continuous and discrete variables are shown as mean ± SD and counts (percentage), respectively. Figures below those statistics denote 95% confidence intervals (lower limit to upper limit).

**Table 2 jcm-11-03864-t002:** Correlation matrix between the changes in the markers of electrochemical antioxidant capacity and anthropometric, hormonal, metabolic, and ferrokinetic/hematimetric variables during the run-in period.

	Q1 (μC)	Q2 (μC)	QT (μC)
*Body mass index (*kg/m^2^*)*	r: −0.086	r: −0.129	r: −0.141
	*p* = 0.638	*p* = 0.481	*p* = 0.441
*Waist circumference (*cm*)*	r: 0.086	r: 0.136	r: 0.196
	*p* = 0.274	*p* = 0.465	*p* = 0.290
*Fat mass (*%*)*	r: −0.190	r: −0.155	r: −0.192
	*p* = 0.298	*p* = 0.396	*p* = 0.293
*Free testosterone (*pmol/L*)*	r: 0.170	r: 0.169	r: 0.156
	*p* = 0.352	*p* = 0.355	*p* = 0.394
*Dehydroepiandrosterone-sulphate (*μmol/L*)*	r: 0.004	r: 0.051	r: 0.048
	*p* = 0.983	*p* = 0.781	*p* = 0.795
*Fasting glucose (*mmol/L*)*	r: −0.158	r: −0.152	r: −0.182
	*p* = 0.389	*p* = 0.408	*p* = 0.318
*Fasting insulin (*pmol/L*)*	r: −0.296	r: −0.287	r: −0.328
	*p* = 0.106	*p* = 0.117	*p* = 0.072
*Homeostasis model assessment of insulin resistance*	r: −0.339	r: −0.335	**r: −0.374**
	*p* = 0.062	*p* = 0.065	** *p* ** ** = 0.038**
*Ferritin (*μg/L*)*	r: 0.116	r: 0.023	r: 0.100
	*p* = 0.528	*p* = 0.901	*p* = 0.586
*Total iron binding capacity (*μg/dL*)*	r: −0.345	r: −0.216	r: −0.302
	*p* = 0.053	*p* = 0.236	*p* = 0.093
*Transferrin Saturation (*%*)*	r: −0.154	**r: −0.416**	r: −0.262
	*p* = 0.401	** *p* ** ** = 0.018**	*p* = 0.147
*Plasma hemoglobin (*g/L*)*	**r: −0.401**	r: −0.229	r: −0.340
	** *p* ** ** = 0.023**	*p* = 0.207	*p* = 0.057

We estimated the changes in each variable as areas under the curve. Values in boldface reached statistical significance. Abbreviations, r: Pearson’s correlation coefficient.

## Data Availability

All data sets generated during and/or analyzed during the current study are not publicly available but are available from the corresponding author on reasonable request.

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
