# Peer review of "Effect of Iron Depletion by Bloodletting vs. Observation on Oxidative Stress Biomarkers of Women with Functional Hyperandrogenism Taking a Combined Oral Contraceptive: A Randomized Clinical Trial"

_jcm, 2022, doi:10.3390/jcm11133864_

Round 1

Reviewer 1 Report

The manuscript by Luque-Ramírez et al. examines the ability of bloodletting to prevent the oxidative stress caused by Combined oral contraceptives (COC) in patients with functional hyperandrogenism. The study seems very interesting because oxidative stress is increased in functional hyperandrogenism patients undergoing COC therapy. The study incrementally adds to their previously published articles suggesting that bloodletting doesn’t influence hypertension or insulin sensitivity; this study utilizes the same samples for assessing oxidative stress. Like their previous findings, the authors did not find any effect of bloodletting on oxidative stress in these subjects, although the ferritin levels were reduced after bloodletting.

 One of the significant strengths of this work is that the study is a randomized clinical trial. Although the mitochondrially generated ROS is the major contributor to oxidative stress, the authors seem to have discounted the other sources of ROS generation in their introduction. Introducing the ROS generated through different sources and discussing it would provide more impactful information to the researchers.

As the first article from this study was published in 2020, It is not clear if the assays were performed on the fresh sample or the stored samples, as sample storage has been known to influence oxidative stress markers.

A better description of Table 2 is needed. The data can also be presented like Figure 2 from this (“Bloodletting has no effect on the blood pressure abnormalities of hyperandrogenic women taking oral contraceptives in a randomized clinical trial”) article from the same trial. Figure 3 doesn’t have a run-in period.  

They could also add to the limitations that the TBARS assay used for lipid peroxidation is sometimes non-specific. 

Author Response

RESPONSE TO REVIEWER 1
The manuscript by Luque-Ramírez et al. examines the ability of bloodletting to prevent the oxidative stress caused by Combined oral contraceptives (COC) in patients with functional hyperandrogenism. The study seems very interesting because oxidative stress is increased in functional hyperandrogenism patients undergoing COC therapy. The study incrementally adds to their previously published articles suggesting that bloodletting doesn’t influence hypertension or
insulin sensitivity; this study utilizes the same samples for assessing oxidative stress. Like their previous findings, the authors did not find any effect of bloodletting on oxidative stress in these subjects, although the ferritin levels were reduced after bloodletting.
ANSWER: We really thank you for your excellent job in reviewing our manuscript. We have revised our manuscript in depth according to your insights. Please find the response to your specific questions below:

Q1. One of the significant strengths of this work is that the study is a randomized clinical trial. Although the mitochondrially generated ROS is the major contributor to oxidative stress, the authors seem to have discounted the other sources of ROS generation in their introduction. Introducing the ROS generated through different sources and discussing it would provide more impactful information to the researchers.
ANSWER: Thank you for your suggestion. Accordingly, we have expanded this subject in the revised Introduction. Pages: 1-2.

Q2. As the first article from this study was published in 2020, It is not clear if the assays were performed on the fresh sample or the stored samples, as sample storage has been known to influence oxidative stress markers.
ANSWER: Regarding your query, the prospective longitudinal study design did not allow us to conduct oxidative stress biomarker analyses on fresh samples. As pointed out, sample storage can influence on oxidative stress marker measurements. To minimize this impact, all samples followed the same strict storage protocol: i) we separated specific aliquots that were immediately frozen at
−80ºC, and ii) those frozen aliquots were not thawed in any case until their assay for oxidative stress markers was conducted. These storing conditions appeared to be optimal to determine antioxidants in other biological fluids (doi: 10.1016/j.rvsc.2020.10.027). In addition, all measurements were performed in duplicate, using kits from the same lot provided by the manufacturer in order to minimize interassay variability. Anyhow, we have added this point as a study limitation (Page 13), since we cannot rule completely out some impact of the storage conditions on our findings.

Q3. A better description of Table 2 is needed. The data can also be presented like Figure 2 from this (“Bloodletting has no effect on the blood pressure abnormalities of hyperandrogenic women taking oral contraceptives in a randomized clinical trial”) article from the same trial. Figure 3 doesn’t have a run-in period.
ANSWER: Data displayed in the Table 2 are now shown as Figure 3. The previous Figure 3 (Figure 4 in the new version) does not include a run-in period because of changes during that phase of the study are shown in Figure 2.

Q4. They could also add to the limitations that the TBARS assay used for lipid peroxidation is sometimes non-specific.
ANSWER: Thank you. You are right. We have added that limitation of the study paragraph to the appropriate (Page 13; reference 59).

Once again, we truly thank you for your review. We really hope that we have properly addressed all your concerns about our work.

Reviewer 2 Report

In the present manuscript, the study design and methods were used appropriate for the aim of the study. All experimental methods were appropriately selected and adequately described in this manuscript. Data analysis was performed using the suitable statistical methods. The tabular materials and figures facilitate the understanding of results obtained by authors. Therefore, the data in this study are clearly presented. All of the discussion is relevant and very interesting. In addition, both the discussion and conclusion are balanced and adequately supported by the date.

The manuscript is divided into subsections in four subheadings: Introduction, Material and Methods, Results and Discussion so it facilitates reading this article and corresponds to the scheme of research article. The title and abstract describe the essential aspects of the investigation. The title of the article is appropriate and informative. The abstract is concise and in the correct form.

Author Response

RESPONSE TO REVIEWER 2

We really appreciate the detailed review of our manuscript, as well as your comments. Following your suggestion, we have performed a spell checking.